

Seasonal prediction skill of East Asian summer monsoon in CMIP5-Models
**Bo Huang,\* Ulrich Cubasch, Christopher Kadow**
**Institute of Meteorology, Freie Universität Berlin,**
**Carl-Heinrich-Becker-Weg 6-10, 12165 Berlin, Germany**
**Email: huangb@zedat.fu-berlin.de**
**ABSTRACT**
The East Asian summer monsoon (EASM) is an important part of the global climate system
and plays a vital role in the Asian climate. Its sub-seasonal-to-seasonal predictability is a
long-standing issue within the monsoon scientist community. In this study, we analyse the
seasonal (with six months lead time) prediction skill of the EASM rainfall and its associated
general circulation in non-initialised and initialised simulations for the years 1979-2005
performed by six prediction systems (*i.e.*, the BCC-CSM1-1, the CanCM4, the GFDL-
CM2p1, the HadCM3, the MIROC5 and the MPI-ESM-LR) from the Coupled Model
Intercomparison Project phase 5 (CMIP 5). We find that the simulation of the zonal wind is
significantly improved in initialised simulations compared to non-initialized simulations.
Based on the knowledge that zonal wind indices can be used as potential predictors for the
EASM, we selected an EASM index based upon the zonal wind for further analysis. The
assessment show that the GFDL-CM2p1 and the MIROC5 add prediction skill in simulating
the EASM index with initialisation, the BCC-CSM1-1, the CanCM4, and the MPI-ESM-LR
change the skill insignificantly, and the HadCM3 indicates a decreased skill score. The
different response to the initialisation can be traced back to the ability of the models to
capture the ENSO (El Niño-Southern Oscillation)-EASM coupled mode, particularly the
Southern Oscillation-EASM coupled mode. In summary, we find that the GFDL-CM2p1 and
the MIROC5 are capable to predict the EASM on a seasonal time-scale after initialisation.
**Key Words**: East Asian summer monsoon; initialisation; seasonal prediction; ENSO-EASM
coupled mode; CMIP5





## 1.  INTRODUCTION

The Asian monsoon is the most powerful monsoon system in the world due to the thermal contrast between the Eurasian continent and the Indo-Pacific Ocean. Its evolution and variability critically influences the livelihood and the socio-economic status of over two billion residents who live in the Asian monsoon dominated region. It encompasses two sub-monsoon systems (e.g., South Asian monsoon-SAM and East Asian monsoon-EAM; Wang, 2006). In summer time (June-July-August), the EAM, namely, the East Asian summer monsoon (EASM) occurs from the Indo-China peninsula to the Korean Peninsula and Japan, and shows strong intraseasonal-to-interdecadal variability (Ding and Chan, 2005). Thus, accurate predictions of the EASM is an important and long-standing issue in climate science.

To predict the EASM, there are two approaches, statistical prediction and dynamical prediction, respectively. The statistical method seeks the relationship between the EASM and a strong climate signal (e.g., ENSO, NAO; Wang et al., 2015;Wu et al., 2009;Yim et al., 2014). This method is limited by the strength of the climate signal. The other method is dynamical prediction. It employs climate model to predict the EASM (Kang and Yoo, 2006;Lee et al., 2010;Sperber et al., 2001;Wang et al., 2008a;Yang et al., 2008;Kim et al., 2012). Two kinds of climate models have been developed in the past few decades, atmosphere general circulation model (AGCM) and coupled atmosphere-ocean general circulation model (AOGCM). Both the two kinds of model have been used to predict the EASM (Kang et al., 2004;Wang et al., 2005;Wang et al., 2007;Wang et al., 2008a;Zhou et al., 2009). For AGCMs, the lower boundary conditions (*e.g.* SST: sea surface temperature) is required. An external ocean model is applied to predict the SST. Then the prescribed SST is employed as the lower boundary conditions to force the AGCMs. However, this method shows a low prediction skill over East Asia, especially in monsoon season (Wang et al., 2005;Barnston et al., 2010), because the AGCMs fail to produce the realistic SST-rainfall relationships in monsoon season (Wang et al., 2005;Wang et al., 2004). Therefore, the monsoon community endeavours to predict the EASM with AOGCMs (Zhou et al., 2009;Wang et al., 2008a;Jiang et al., 2013;Kim et al., 2012).

AOGCMs have proved to be the most valuable tools in predicting the EASM (Zhou et al., 2009;Wang et al., 2008a;Jiang et al., 2013;Kim et al., 2012). However, the performance of AOGCMs in predicting the EASM on seasonal time-scale strongly depends on their ability to reproduce the teleconnection between EASM and SST (Sperber et al., 2001) and the initialisation (Wang et al., 2005).  In the coupled model inter-comparison project (CMIP)





phase 3 (CMIP3; Meehl et al., 2007) era, the models simulate not only a too weak SST-
monsoon teleconnection (Kim et al., 2008;Kim et al., 2011), but also a too weak East Asian
zonal wind-rainfall teleconnection (Sperber et al., 2013). Compared to CMIP3 models, CMIP
phase 5 (CMIP5; Taylor et al., 2012) models improve the representation of monsoon status
(Sperber et al., 2013). Therefore, given initial conditions, the CMIP5 models have potential to
predict the EASM.
Initial conditions play a vital factor in predicting the EASM on sub-seasonal to
seasonal time-scale (Kang and Shukla, 2006). Under current set up of initialisation, the
CMIP5 models show the ability to predict the SST indicator (*i.e.,* El Niño-Southern
Oscillation-ENSO index) up to 15 months in advance (Choi et al., 2016;Meehl et al.,
2014;Meehl and Teng, 2012). This extended prediction skill of the ENSO suggests that the
EASM can be predicted on a seasonal time-scale if the dynamic link between the ENSO and
monsoon circulations is well represented in these models. Two scientific questions will be
addressed in this study: 1. How realistic are the initialized CMIP5 models in representing the
EASM? 2. To what extend reproduce the model's teleconnection between the ENSO and the
EASM?
In this paper, we inter-compare the influence of the initialisation on the capability of
the CMIP5 model to capture the EASM and the ENSO-EASM teleconnections. The model
simulations, comparison data and methods are introduced in Section 2. Section 3 describes
the seasonal skill of the rainfall predictions and the prediction of the associated general
circulation of the EASM. The mechanism causing the differential response of the models to
the initialisation is presented in Section 4. The main conclusions and discussions are
summarised in Section 5.
**2.    MODELS, DATA AND METHODS**
**2.1 MODELS AND INITIALISATION**
In this study, we assess six prediction systems which have contributed to CMIP5 in historical
and decadal hindcast simulations (Table 1). We employ these six prediction systems of
CMIP5 in our study which have performed a yearly initialisation. Only these systems provide
data to study the effect of initialisation on seasonal time-scale. The BCC-CSM1-1 has three
ensemble members which are initialised on 1st September, 1st November and 1st January,
respectively. The initialisation of HadCM3 takes place on each pre-year 1st November while
the other four systems are initialised on 1st January. The full-field initialisation is named
HadCM3-ff to distinguish it from the anomaly initialisation in HadCM3. Because of spatial





coverage of the precipitation observations, we select the satellite era (1979 to 2005) for our
study. The first lead year results from initialised simulations are used to assess the seasonal
predicting skills of the CMIP5 models. The initialisation strategies of all modelling groups
from CMIP5 decadal prediction experiments have been summarised in Meehl *et al.* (2014).
The brief configurations of the six prediction systems are presented in Table 2.
**2.2 COMPARISON DATA**
The main datasets which are used for the comparison in this study include: (1)
monthly precipitation data from the Global Precipitation Climatology Project (GPCP; Adler
et al., 2003); (2) monthly circulation data from ECMWF Interim re-analysis (ERA-Interim;
Dee et al., 2011); and (3) monthly mean SST from National Oceanic and Atmospheric
Administration (NOAA) improved Extended Reconstructed SST version 4 (ERSST v4;
Huang et al., 2015). All the model data and the comparison data are remapped onto a
common grid of 2.5°x2.5° by bi-linear interpolation to reduce the uncertainty induced by
different data resolutions.
**2.3 METHODS**
We apply the pattern correlation coefficient (PCC) to analyse the model performance
in capturing the spatial pattern with reference to the observational data. It is the Pearson
product-moment coefficient of linear correlation between a single variables on two different
spatial patterns (Barnett and Schlesinger, 1987). There are two types of pattern correlation
statistics: centred and un-centred. The centred (un-centred) statistic measures the similarity of
two patterns after (without) the removal of the global mean. We choose the un-centred PCC
in our study due to the fact that centred correlations alone are not sufficient for the attribution
of seasonal prediction (Mitchell et al., 2001). The un-centred PCC is defined by:

$$PCC = \frac{\sum_{x=1}^{n} \sum_{y=1}^{m} w_{(x,y)} F_{(x,y)} A_{(x,y)}}{\sqrt{\sum_{x=1}^{n} \sum_{y=1}^{m} w_{(x,y)} F_{(x,y)}^2 \sum_{x=1}^{n} \sum_{y=1}^{m} w_{(x,y)} A_{(x,y)}^2}}$$


where n and m are grids on longitude and latitude, respectively. $F_{(x,y)}$ and $A_{(x,y)}$ represent two
dimensions comparison and validating value. $w_{(x,y)}$ indicates the weighting coefficient for
each grid. An equal weighting coefficient was applied due to the study area is East Asia
where we can omit the convergence of the longitudes with the latitudes
We also employ the anomaly correlation coefficient (ACC) to analyse the model
performance in reproducing observational variations. The ACC is the correlation between



anomalies of forecasts and those of verifying values with the reference values, such as climatological values (Drosdowsky and Zhang, 2003). Its definition is:

$$ACC = \frac{\sum_{i=1}^{n} w_i (f_i - \bar{f})(a_i - \bar{a})}{\sqrt{\sum_{i=1}^{n} w_i (f_i - \bar{f})^2 \sum_{i=1}^{n} w_i (a_i - \bar{a})^2}}, (-1 \leq ACC \leq 1)$$

$$f_i = F_i - C_i, \bar{f} = \left( \sum_{i=1}^{n} w_i f_i \right) \Big/ \sum_{i=1}^{n} w_i$$

$$a_i = A_i - C_i, \bar{a} = \left( \sum_{i=1}^{n} w_i a_i \right) \Big/ \sum_{i=1}^{n} w_i$$

where $n$ is the number of number of samples, and $F_i$, $A_i$, $C_i$ represent comparison, verifying value, and reference value such as climatological value, respectively. Also, $\bar{f}$ is the mean of $f_i$, $\bar{a}$ is the mean of $a_i$, and $w_i$ indicates the weighting coefficient. If the variation of anomalies of comparison dataset is perfectly coincident with that of the anomalies of verifying value, ACC will take 1 (the maximum value). Otherwise, if the variation is completely reversed, ACC is -1 (the minimum value).

The root-mean-square-error (RMSE) is employed to check the model deviation from the observation and its definition is:

$$RMSE = \sqrt{\sum_{i=1}^{n} w_i D_i^2} \Big/ \sqrt{\sum_{i=1}^{n} w_i}$$

where $D_i$ represents the deviation between comparison and verifying value, $w_i$ is the weighting coefficient for each sample, and n is the number of samples. If RMSE is closer to zero, it means that the comparisons are closer to the verifying values.

3.    **SEASONAL PREDICTION SKILL OF THE EASM**

The EASM has complex spatial and temporal structures that encompass the tropics, subtropics, and midlatitudes (Tao and Chen, 1987; Ding, 1994). In late spring, an enhanced rainfall pattern is observed in the Indochina Peninsula and in South China Sea. Then, the rainfall belt advances northwards to the south of China. In early summer, the rainfall concentration occurs in the Yangtze River Basin and in southern Japan, namely, the Meiyu



and Baiu season, respectively. The rainfall belt can reach as far as northern China, the Korean
Peninsula (called the Changma rainy season) and central Japan in July (Ding, 2004;Ding and
Chan, 2005).
The EASM is characterised by both seasonal heterogeneous rainfall distribution and
associated large-scale circulation systems (Wang et al., 2008b). In the summer season, water
moisture migrates from the Pacific Ocean to central and eastern Asia, which is carried by the
southwest surface winds. Generally, a strong summer monsoon year is followed by
precipitation in northern China, while a weak summer monsoon year is usually accompanied
by heavier rainfall along the Yangtze River basin (Zhou and Yu, 2005;Ding, 1994).
The prediction skill of the EASM rainfall and the associated general circulation
variable (*i.e.*, zonal and meridional wind, and mean sea level pressure) is presented in Figure
1. These variables are resource of monsoon index (Wang et al., 2008b). Table 3 shows the
contribution of these variables in the EASM. Their abbreviations follow the guideline of
CMIP5 (Taylor et al., 2012). Without initialisation, the models show an acceptable
performance in capturing the observed spatial variation (with high PCC) of the six variables,
but a poor performance in simulating their temporal variation (with low ACC). After
initialisation, we can see that the models show a higher ACC of the six variables. However,
there is no improvement in simulating the spatial variation (PCC). The improvement of
simulating the temporal variation of zonal winds (*i.e.*, ua850 and ua200) is larger than of the
rainfall and meridional winds. One can exploit this improvement by using, a general
circulation based monsoon index as a tool to predict the EASM.
In the recent decades, more than 25 general circulation indices have been produced to
research the variability and long-term change of the EASM. Wang *et al.* (2008) classified
them into five categories and discussed their ability to capture the main features of the
EASM. They found that the Wang and Fan index (hereafter WF-index; 1999) shows the best
performance in capturing the total variance of the precipitation and three-dimensional
circulation over East Asia. We, thus, select the WF-index for the further analysis. Its
definition is standardised average zonal wind at 850 hPa in (5°-15ºN, 90°-130ºE) minus in
(22.5°-32.5ºN, 110°-140ºE). The WF-index is a shear vorticity index which often described
by a north-south gradient of the zonal winds. In positive (negative) phase of the WF-index
years, two strong (weak) rainfall belts locate at the Indo China Peninsula-to-the Philippine
Sea and the northern China-to-the Japan Sea, and a weak (strong) rainfall belt occurs from the
Yangtze river basin-to-the south of Japan.



In the non-initialised simulations, none of the models captures the observed EASM, as
indicated by an insignificant ACC (Figure 2). The CanCM4 and the GFDL-CM2p1 simulate
a negative phase, while the BCC-CSM1-1, the HadCM3, the MIROC5 and the MPI-ESM-LR
all predict a positive phase of the EASM. After initialisation, the CanCM4, the GFDL-
CM2p1 and the MIROC5 improve the skill to simulate the EASM, the MPI-ESM-LR
displays hardly any reaction, while the BCC-CSM1-1 and the HadCM3 shows a worse
performance than before. Particularly with anomaly initialisation, the HadCM3 significantly
loses its prediction skill in capturing the EASM.
**4.     EASM-ENSO COUPLED MODE IN CMIP5**
We employ the EOF method to analyse the leading EOF modes of six meteorological
variables anomaly in the EASM region (0°-50ºN, 100°-140ºE). The first EOF mode of
rainfall is characterised by a "sandwich" pattern which shows sharp contrast between the
prominent rainfall centre over Malaysia, the Yangtze River valley and the south of Japan, and
the enhanced rainfall over the Indo-China Peninsula and the Philippine Sea. The increased
precipitation is associated with cyclones in the low-level (850 hPa) and anti-cyclones in the
upper level (200 hPa) (not shown).
The correlation coefficient of the first eigenvector and the associated principal
component (PC) between the model simulation and the observation in the non-initialised and
the initialised simulation is presented in Figure 3. The models can capture the eigenvector of
the first EOF for the six meteorological fields in non-initialised simulation. However, they
fail to reproduce the associated PC of the first leading EOF mode. Compared to the non-
initialised simulation, the models show no improvement to simulate the first leading EOF
mode of rainfall, but exhibit a better performance in representing the first leading EOF mode
of zonal wind. The CanCM4 and the GFDL-CM2p1 capture the first PC of ua850, but not the
other five models. For the zonal wind at 200 hPa, the BCC-CSM1-1 fails to simulate its first
EOF mode while the other six models can. Then, only the GFDL-CM2p1 accurately
simulates the first EOF eigenvectors and the associated PC of va850, which cannot be
reproduced in the other models. None of the models captures the spatial-temporal variation of
the first EOF mode of meridional wind at 200 hPa. In addition, the GFDL-CM2p1 and the
MIROC5 simulates a reasonable leading EOF mode and associated PC of psl, while the other
models do not capture it.
Figure 4 shows the fractional (percentage) variances of the six variables of the first
EOF mode with the total variances from the observation, and the model simulation in non-





initialisation and in initialisation. The observational total variances for the pr, the ua850, the
ua200, the va850, the va200 and the psl, are depicted by the first lead EOF mode in 21.2,
59.0, 36.5, 20.6, 28.5 and 50.0 percent, respectively. The models simulate the comparable
explanatory variances, which show a slight discrepancy for the first leading mode in the non-
initialisation. From non-initialised simulation to initialised simulation, the CGCMs tend to
enhance the first EOF lead mode due to the fact that they show larger fractional variances of
the total variances of the six variables. We note that the CanCM4 and the GFDL-CM2p1
significantly increase the fractional variances from non-initialisation to initialisation.

The ENSO is a dominant mode of the inter-annual variability of the coupled ocean

and atmosphere climate system, which has strong effects on the inter-annual variation of the
EASM (Wu et al., 2003;Wang et al., 2000). Wang et al. (2015) summarised the first EOF
lead mode of the ASM is the ENSO developing mode. As previously mentioned, the first
EOF mode is improved in the initialised simulations, compared to the non-initialised
simulation. This also can be found in the ENSO indices (Figure 5). Niño3.4 is calculated by
the SST anomaly in the central Pacific (190-240ºE, 5ºS-5ºN), while the southern oscillation
index (SOI) is based upon the anomaly of the sea level pressure differences between Tahiti
(210.75ºE, 17.6ºS) and Darwin (130.83ºE, 12.5ºS). To calculate the SOI, we interpolate the
grid data to the Tahiti and the Darwin point by bilinear interpolation.

The individual members and their ensemble mean of the six models show a low

correlation coefficient to the observational Niño3.4 and the SOI in the non-initialised
simulations. Niño3.4 and SOI represent the oscillation of two components in the earth
system, the ocean and the atmosphere, respectively. These two indices show strong anti-
phase in the observation, with correlation range is -0.94 to -0.92 for four seasons (DJF,
MAM, JJA, SON; Figure 5). The models describe the anti-correlation between Niño3.4 and
the SOI, but weaker than observed. Compared to the non-initialisation, there is a significant
improvement for models in capturing the observational Niño3.4 and the SOI after
initialisation. Initialisation lowers the spread of ensemble members in predicting Niño3.4 and
the SOI in all the six models. However, initialisation does not prominently change the
correlation between Niño3.4 and the SOI in the model simulations. With initialisation, the
GFDL model shows a weaker correlation between Niño3.4 and the SOI, while the HadCM3
models illustrate a stronger correlation. It is worth mentioning that after initialisation the
ensemble mean of each model outperforms its individual members in capturing Niño3.4 and
the SOI. The correlation coefficient between Niño3.4 and the SOI of MME is ~0.8 in both
non-initialised and initialised simulations.

The EASM strongly relies on the pre-seasons ENSO signal due to the lag response of

the atmosphere to the SST anomaly (Wu et al., 2003). The lead-lag correlation coefficients
between the EASM index and the Niño3.4, and the SOI from JJA(-1) to JJA(+1) are
illustrated in Figure 6. The pre-season Niño3.4 (SOI) presents a significant negative
(positive) correlation to the EASM, while the post-season Niño3.4 (SOI) shows a notable
positive (negative) correlation. This lead-lag correlation coefficient phase is called the
Niño3.4-/SOI-EASM coupled mode (Wang et al., 2008b). In the non-initialised cases, the
models do not produce the teleconnection between the ENSO and the EASM. The CanCM4,
the HadCM3 and the MPI-ESM-LR fail to represent the lead-lag correlation coefficient
difference between pre-/post-season ENSO and EASM. The BCC-CSM1-1, the GFDL-
CM2p1 and the MIROC5 capture the coupled mode of the ENSO and the EASM. However,
the pre-season ENSO has a weak effect on the EASM. Compared to the non-initialised cases,
the MIROC5 and the GFDL-CM2p1 both demonstrate a significant improvement in
simulating Niño3.4 (SOI)-EASM coupled mode in the initialisation. The BCC-CSM1-1, the
HadCM3, and the HadCM3-ff show no improvement, with insignificant correlation between
Niño3.4 (SOI) and the EASM. The CanCM4 and the MPI-ESM-LR indicate a higher
correlation between the EASM and the simultaneous-to-post-season ENSO than to the pre-
season ENSO.
**5.    SUMMARY AND DISCUSSION**
Six earth system models from CMIP5 have been selected in our study. We have analysed the
improvement of the rainfall, the mean sea level pressure, the zonal wind and the meridional
wind in the EASM region from non-initialisation to initialisation. The low prediction skill of
the summer monsoon precipitation is due to the uncertainties of cloud physics and cumulus
parameterisations in the models (Lee et al., 2010;Seo et al., 2015). The models show a better
performance in capturing the inter-annual variability of zonal wind than the precipitation after
initialisation (Figure 1). Thus, the zonal wind index is an additional factor which can indicate
the prediction skill of the model. When, we calculate the WF-index in both non-initialised
and initialised simulations, the GFDL-CM2p1 and the MIROC5 show a significant
advancement in simulating the EASM from non-initialised to initialised simulation with a
lower RMSE and a higher ACC (Figure 2). There is only a slight change in the WF calculated
from the BCC-CSM1-1, the CanCM4 and the MPI-ESM-LR data after initialisation.



Compared to the non-initialised simulation, the HadCM3 loses prediction skill, especially
with anomaly initialisation.

To test the possible mechanisms of the models' performance in the non-initialisation

and the initialisation, we have calculated the leading mode of the six fields which are
associated to the EASM. The models demonstrate a better agreement with the observational
first EOF mode in the initialised simulations (Figure 3). The first lead mode of zonal wind at
200 hPa shows a significant improvement in the models except the BCC-CSM1-1 with
initialisation. Therefore, a potential predictor might be an index based upon the zonal wind at
200 hPa. Compared to the non-initialisation, the models enhance the first EOF mode with a
higher fraction of variance to the total variance after initialisation (Figure 4). The first EOF
mode of the EASM is the ENSO developing mode (Wang et al., 2015). We have analysed the
seasonal simulating skill of Niño3.4 and the SOI in each model (Figure 5). The models show
a poor performance in representing Niño3.4 and the SOI in the non-initialised simulation.
Initialisation improves the model simulating skill of Niño3.4 and the SOI. The initialised
simulations decrease the spread of ensemble members in the models. We found that there is
no significant change in the models reproducing the correlation between Niño3.4 and the SOI
from non-initialisation to initialisation.

In general, the pre-season warm phase of the ENSO (El Niño) leads to a weak EASM

producing more rainfall over the South China Sea and northwest China, and less rainfall over
the Yangtze River Valley and the southern Japan; the cold phase of the ENSO (La Niña)
illustrates a reverse rainfall pattern to El Niño in East Asia. The pre-season Niño3.4 (SOI)
exhibits a strong negative (positive) correlation to the EASM, while the correlation between
the post-season Niño3.4 (SOI) and the EASM illustrates an anti-phase as the pre-season
(Figure 6). In the non-initialised simulations, the models do not capture Niño3.4-/SOI-EASM
coupled mode. We found that only the MIROC5 has the ability to represent the Niño3.4-
EASM coupled mode with initialisation. For the SOI-EASM coupled mode, the GFDL-
CM2p1 and the MIROC5 capture it in the initialisation, while the BCC-CSM1-1, the
HadCM3, the HadCM2-ff, the CanCM4 and the MPI-ESM-LR do not.

The model exhibits a better performance in simulating the general circulation of the

EASM with initialisation. Thus, initialisation is helpful in forecasting the EASM on a
seasonal time-scale. There are two initialisation methods in our study, full-field initialisation
and anomaly initialisation (Table 1). The full-field initialisation produces more skilful
predictions on the seasonal time-scale in predicting regional temperature and precipitation



(Magnusson et al., 2013;Smith et al., 2013). But, for predicting the EASM, there is no
significant difference between the two methods. We can see that both the GFDL-CM2p1 and
the MIROC5 have a significant improvement in capturing the EASM, with full-field and
anomaly initialisation, respectively. Only the HadCM3 was initialised by the two
initialisation techniques. However, both these two initialised techniques are producing poor
predictions of the EASM with no major differences.
The initialisation strategy of the models is to initialise with the observed atmospheric
component (*i.e.*, zonal and meridional wind, geopotential height, *etc.*) and the SST (Meehl et
al., 2009;Meehl et al., 2014;Taylor et al., 2012). With initialisation, the SST conveys its
information via the large heat content of ocean to the coupled system. Therefore, an index
indicating an ocean oscillation like Niño3.4 shows a seasonal-to-decadal prediction skill
(Choi et al., 2016;Luo et al., 2008;Jin et al., 2008). The models studied here demonstrate a
prediction skill in simulating Niño3.4 and the SOI due to this effect. The change of the
correlation between Niño3.4 and the SOI is insignificant from non-initialised to initialised
simulations. We therefore conclude that the relationship between Niño3.4 and the SOI
depends more on the model parameterisation than on the initial condition.
Wang *et al*. (2015) found that the second EOF mode of ASM is the Indo-western
Pacific monsoon-ocean coupled mode, the third is the Indian Ocean dipole (IOD) mode, and
the fourth is trend mode. The Indo-western Pacific monsoon-ocean coupled mode is the
atmosphere-ocean interaction mode (Xiang et al., 2013;Wang et al., 2013), which is
supported by positive thermodynamic feedback between the western North Pacific (WNP)
anticyclone and the underlying Indo-Pacific sea surface temperature anomaly dipole over the
warm pool (Wang et al., 2015). The IOD increases the precipitation from the South Asian
subcontinent to south-eastern China and suppresses the precipitation over the WNP (Wang et
al., 2015). It affects the Asian monsoon by the meridional asymmetry of the monsoonal
easterly shear during the boreal summer, which can particularly strengthen the northern
branch of the Rossby wave response to the south-eastern Indian Ocean SST cooling, leading
to an intensified monsoon flow as well as an intensified convection (Wang and Xie,
1996;Wang et al., 2003;Xiang et al., 2011;Wang et al., 2015). We noted that the models
simulate a reasonable first EOF mode (Figure 3), but illustrate no skill in capturing the other
EOF leading modes (not shown). We argue that the models cannot well represent the
monsoon-ocean interaction, even with initialisation. Then, the models do not simulate the
third EOF leading mode of the EASM since the predictability of the IOD extends only over a



three-month time-scale (Choudhury et al., 2015). The current initialisation strategies (both
anomaly and full-field) enhance the ENSO signal in the model simulations with higher
explained fraction of variance. Kim et al. (2012) described a similar finding in ECMWF
System 4 and NCEP Climate Forecast System version 2 (CFSv2) seasonal prediction
simulations. This overly strong modulation of the EASM by ENSO due to the models well
predict ENSO on seasonal time-scale with initialisation (Kim et al., 2012;Jin et al., 2008).

It is worth mentioning that it was an extremely weak monsoon and strong El Niño

year in 1998. The CanCM4, the GFDL-CM2p1, the MIROC5 and the MPI-ESM-LR have the
ability to simulate the extreme monsoon event, while the BCC-CSM1-1, and the HadCM3 do
not capture it even with initialisation. There is potential for the BCC-CSM and the HadCM
models to improve the teleconnection between the ENSO and the EASM.

This study has discussed six CMIP5 models in predicting the EASM on seasonal

time-scale. The six models are earth system coupled models which present a better SST-
monsoon teleconnection than IRI (International Research Institute for Climate and Society)
models (Barnston et al., 2010) and CMIP3 models (Sperber et al., 2013). The CMIP5 models
show a comparable prediction skill as current seasonal forecast application systems, the
ECMWF System and the NCEP CFS, respectively. Both the two application systems have
low prediction skill of EASM (Jiang et al., 2013;Kim et al., 2012).

We have compared six CMIP5 systems with their respective initialisation strategies.

The GFDL-CM2p1 and the MIROC5 have the potential to serve as seasonal forecast
application system even with their current initialisation method. These models have great
potential to optimise the SST-EASM interaction simulation performance to improve their
seasonal prediction skill of the EASM.

**Acknowledgements**

This work was supported by the China Scholarship Council (CSC) and the Freie Universität
Berlin. We would like to thank the climate modelling groups listed in Table 1 of this paper
for producing and making their model output available. We acknowledge the MiKlip project
funded by the Federal Ministry of Education and Research and the German Climate
Computing Centre (DKRZ) for providing the data services.

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



Table 1. Details of the prediction systems investigated in this study.

| System | Institute | Resolution | | Non-Initialisation | Initialisation | | Reference |
| | | Atmospheric | Oceanic | Members | Members | Type | |
|---|---|---|---|---|---|---|---|
| **BCC-CSM1-1** | Beijing Climate Center, China | T42L26 | 1lonx1.33lat L40 | 3 | 3 | Full-field | Wu *et al.* (2014) |
| **CanCM4** | Canadian Centre for Climate Modelling and Analysis, Canada | T63L35 | 256 x 192 L40 | 10 | 10 | Full-field | Arora *et al.* (2011) |
| **GFDL-CM2p1** | Geophysical Fluid Dynamics Laboratory, USA | N45L24 | 1lon x 0.33-1lat L50 | 10 | 10 | Full-field | Delworth *et al.* (2006) |
| **HadCM3** | Met Office Hadley Centre, UK | N48L19 | 1.25x1.25 L20 | 10 | 10 + 10 | Full-field and Anomaly | Smith *et al.* (2013) |
| **MIROC5** | Atmosphere and Ocean Research Institute, Japan | T85L40 | 256x192 L44 | 5 | 6 | Anomaly | Tatebe *et al.* (2012) |
| **MPI-ESM-LR** | Max Planck Institute for Meteorology, Germany | T63L47 | GR15 L40 | 3 | 3 | Anomaly | Matei *et al.* (2012) |

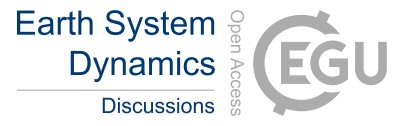


Table 2. Brief summaries of initialisation strategies used by modelling groups in the study. ECMWF: European Centre for Medium-Range Weather Forecasts; GODAS: Global Ocean Data Assimilation System; NCEP: National Centers for Environmental Prediction; S: Salinity; SODA: Simple Ocean Data Assimilation; T: Temperature.

| system | Atmosphere | Ocean | Internet |
|---|---|---|---|
| **BCC-CSM1-1** | - | integration with ocean T nudged to SODA product above 1500 m | http://forecast.bcccsm.ncc-cma.net/ |
| **CanCM4** | ECMWF re-analysis | off-line assimilation of SODA and GODAS subsurface ocean T and S adjusted to reserve model T-S | http://www.cccma.ec.gc.ca/ |
| **GFDL-CM2p1** | GFDL re-analysis | assimilates observations of T, S from World Ocean Database | https://www.gfdl.noaa.gov/multide cadal-prediction-stream/ |
| **HadCM3** | ECMWF re-analysis | off-line ocean re-analysis product | http://cerawww.dkrz.de/WDCC/C MIP5/ |
| **MIROC5** | - | integration using observational gridded ocean T and S | http://amaterasu.ees.hokudai.ac.jp/ |
| **MPI-ESM-LR** | NCEP re-analysis | off-line ocean hindcast forced with NCEP | http://cerawww.dkrz.de/WDCC/C MIP5/ |





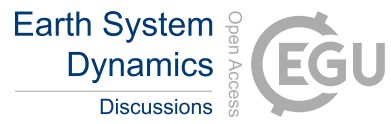


Table 3.  Description of the six variables which contribute to the EASM. The abbreviation of these variables is followed to the guidelines of
CMIP5.

| variable | Standard name | Contribution to the EASM |
|---|---|---|
| pr | Precipitation | Precipitation distribution indicates the strength of EASM |
| psl | Mean sea surface pressure | Differences of mean sea surface pressure between land and ocean lead to EASM |
| ua850 | Zonal winds over 850 hPa | A component of low-level cyclone which transports vapor from ocean to land |
| va850 | Meridional winds over 850 hPa | As ua850, and contributes to Hadley's cell |
| ua200 | Zonal winds over 850 hPa | A component of upper-level Hadley's cell |
| va200 | Meridional winds over 850 hPa | As ua200 |




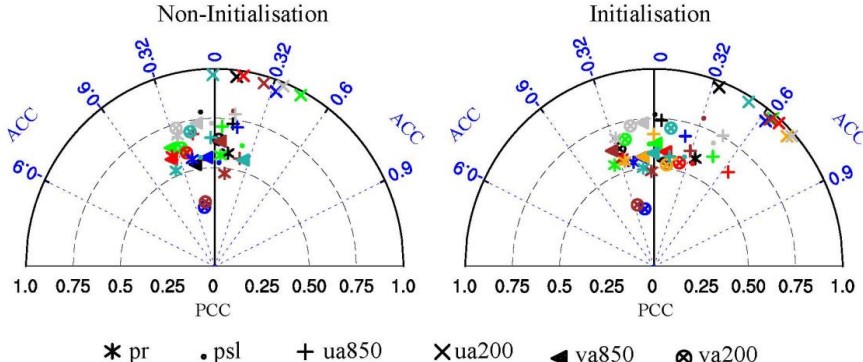


Fig.1. Taylor diagrams display of pattern (PCC) and temporal (ACC) correlation
metrics of six variables between observation and model simulation in the EASM
region (0-50ºN, 100-140ºE). Each coloured marker represents a model, *i.e.*, the BCC-
CSM1-1 (black), the CanCM4 (green), the GFDL-CM2p1 (red), the HadCM3 (blue),
the MIROC5 (brown), the MPI-ESM-LR (light-sea-blue), and the HadCM3-ff
(orange). The GPCP was employed as the reference data for precipitation (*i.e.*, pr)
while wind fields (*i.e.*, ua850, va850, ua200 and va200) and mean sea level pressure
(*i.e.*, psl) were compared by ERA-Interim re-analysis.





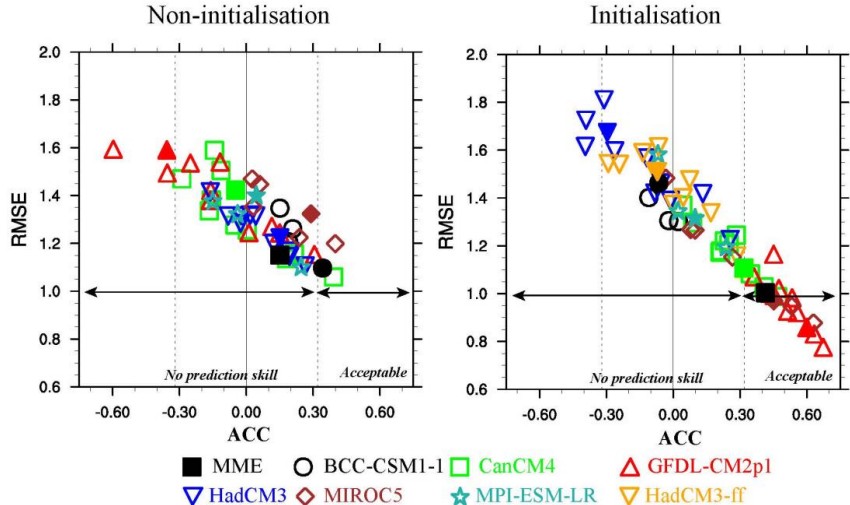


Fig. 2. Performance of the model ensemble member (hollow marker) and its ensemble

mean (solid marker) on the EASM index. The abscissa and ordinates are the anomaly

correlation coefficient (ACC) and the root-mean-square-error (RMSE), respectively.

The observed EASM index is calculated by zonal wind at 850 hPa from the ERA-

Interim re-analysis data. The black dot lines indicate the significant level at 0.1. The

vertical black line represents the correlation between the simulating and the

observational EASM index is 0.








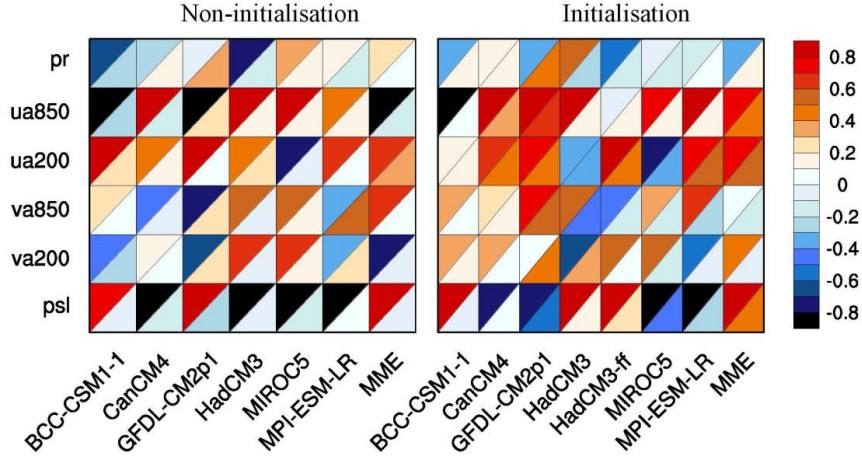


Fig. 3. Portrait diagram display of correlation metrics between the observation and the
model simulation of the first lead EOF mode for the six fields in the non-initialisation
(left) and the initialisation (right). Each grid square is split by a diagonal in order to
show the correlation with respect to both the eigenvector (upper left triangle) and its
associated principal components (lower right triangle) reference data sets.







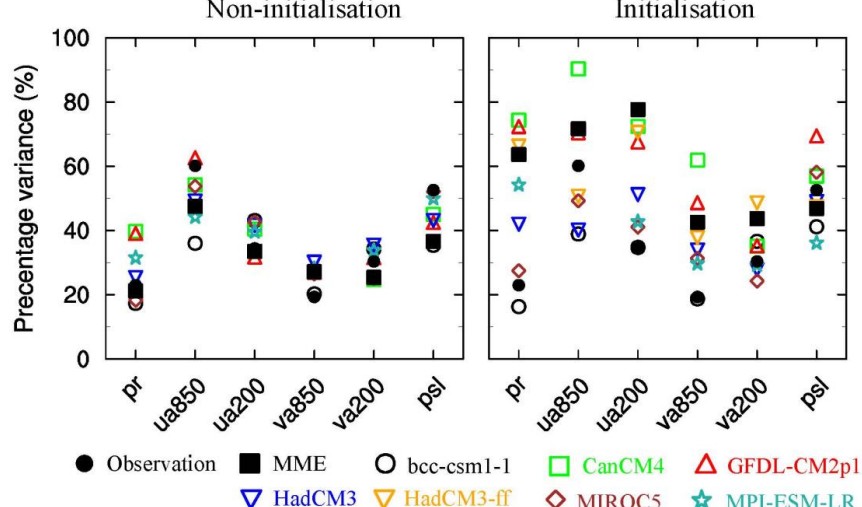


Fig. 4. Fraction variance (%) explained by the first EOF mode for six fields in the
non-initialisation (left) and the initialisation (right).






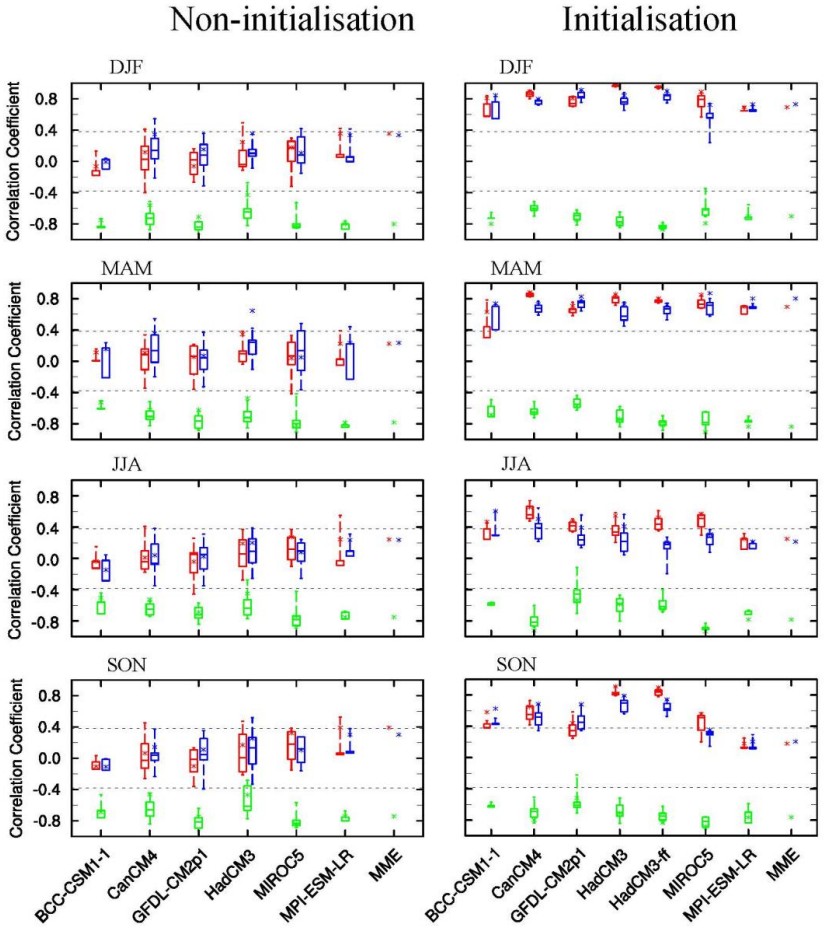


Fig. 5. Model prediction skill in representing the observational Niño3.4 (red), the SOI

(blue) from the DJF to SON in non-initialisation (left) and initialisation (right). Green

diagram shows the correlation coefficient between the model simulated Niño3.4 and

the SOI. Box and whisker diagram shows ensemble mean of each model (asterisk),

median (horizontal line), 25th and 75th percentiles (box), minimum and maximum

(whisker). The two black dotted lines indicate 0.05 significant level based upon

Student's t-test.







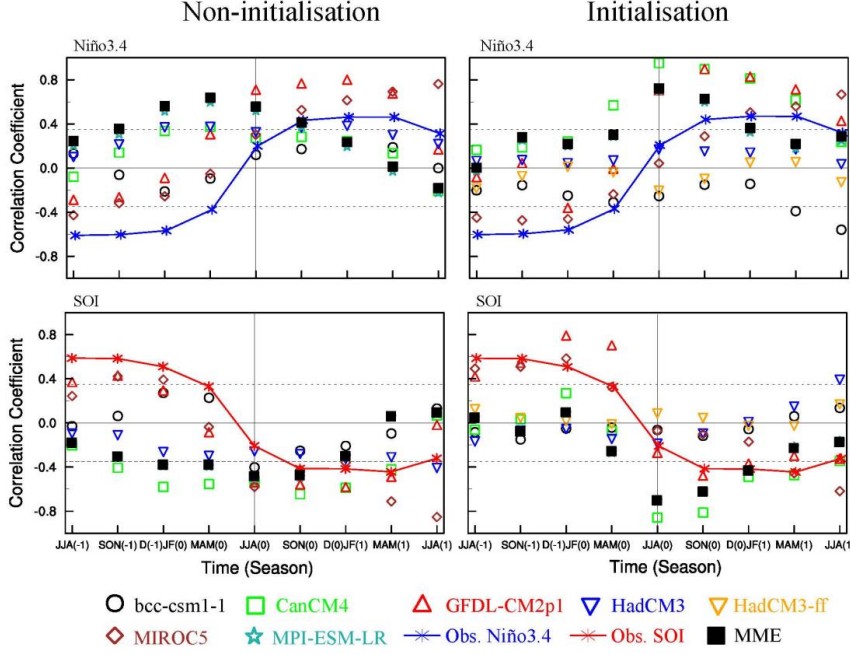


Fig. 6. Lead-lag correlation coefficients between the EASM index and Niño3.4
(upper), and SOI (lower) in non-initialised simulations (left) and initialised ones
(right) for observation (marker line) and models (marker) from JJA(-1) to JJA(+1).
The two black dotted lines are 0.05 significant level based upon Student's t-test. The
vertical line represents JJA(0), where the simultaneous correlations between the
EASM index and Niño3.4, and SOI are shown.