# Peer review of "Seasonal Prediction Skill of East Asian Summer Monsoon in CMIP5-Models"

_Earth System Dynamics, 2017_

## Referee Comment (RC1) · Anonymous Referee #1 · 22 Jun 2017

General comments The manuscript investigated the sub-seasonal-to-seasonal predictability of EASM rainfall and associated six variables in non-initialized and initialized simulation based on the CMIP5 models. Which shows the extend reproduce the model's teleconnection between the EASM and the ENSO. The topic is interesting and study is meaningful to deepen understanding of the EASM predictability and seek a way to improve seasonal EASM prediction. I recommend accept to publish after a major revision. Minor Comments: 1. There are more than 20 models in CMIP5, they are different from each other, the author should clarify the rationality of the six model used in the study. 2. The structures of this paper need to be organized substantially. The authors should pay more attention on the analysis of the performance of EASM with different model instead of introducing the previous results in the main body. 3.

[Figure]

The conclusion of this study is vague and needs to be summarized deeply. Besides, Line327-328 should be moved Section 1 or section 4 Specific Comments: 1. The conclusion of EASM-ENSO coupled mode should be put in the abstract. 2. Line 162-164, the authors should give the sample size in calculating ACC. 3. Apart from the analysis of each figure, the authors should also give a brief summary of each figure. For instance, After Line 187, what is the key findings of Figure 1and 2.

---

## Author Comment (AC1) · 17 Jul 2017

We thank reviewer 1 very much for her/his helpful comments. We took her/his remarks into account and improved the manuscript accordingly.

Minor comments:

1. Comment: There are more than 20 models in CMIP5, they are different from each other, the author should clarify the rationality of the six model used in the study.

Response: You are right. There are more than 20 models contributing to the CMIP5 project. Our manuscript discusses the prediction skill of EASM on seasonal time-scale. Therefore, a yearly initialisation is required. Only the six models have been initialised with a yearly time-frequency. There is a detailed description of the models in section

2.1, line86-96. More information about the six models can be found in Table 1 and 2.

2. Comment: The structures of this paper need to be organized substantially. The authors should pay more attention on the analysis of the performance of EASM with different model instead of introducing the previous results in the main body.

Response: We changed the structure of this paper. The summary (section 6) comes now after the discussion (section 5).

CMIP5 models have been wildly used to investigate the monsoon change in East Asia. In our paper, we solely focus on the seasonal predictability of the EASM aspect. A comprehensive evaluation of the performance of the CMIP5 models exceeds the scope of this paper. Some analysis can be found in Sperber et al. (2013).

Previous study shows that the prediction of the climate on seasonal time-scale is the initial value problem (Meehl et al., 2009). Given suitable initial condition, coupled models have potential to predict the climate. Furthermore, CMIP5 models illustrate significant improvement in simulating the EASM (Sperber et al., 2013). Meanwhile, the models show the ability to predict the SST indicator (i.e., El Niño-Southern Oscillation-ENSO index) up to 15 months in advance (Choi et al., 2016; Meehl et al., 2014; Meehl and Teng, 2012). This extended prediction skill of the ENSO suggests that the EASM can be predicted on a seasonal time-scale if the dynamic link between the ENSO and monsoon circulations is well represented in these models. Analysing the six suitable CMIP5 models, we find that the GFDL-CM2p1 and the MIROC5 add prediction skill in simulating the EASM index with initialisation, the BCC-CSM1-1, the CanCM4, and the MPI-ESM-LR change the skill insignificantly, and the HadCM3 indicates a decreased skill score.

Our paper pursues following structure: why do these models show different reactions? To answer this question, the different response of the models to the initialisation has been evaluated. An EOF method has been employed to analyse the principle components of the models simulated EASM. The result show that the GFDL-CM2p1 and

MIROC5 have better performance to simulate the first EOF leading mode. Wang et al. (2015) found that in EASMs the first EOF leading mode is the ENSO developing mode. We checked the ENSO-EASM coupled mode in the six models investigated, and found that only the GFDL-CM2p1 and MIROC5 simulate this ENSO-EASM coupled mode. This answers the question about the different responses to the initialisation in the models.

3. Comment: The conclusion of this study is vague and needs to the summarized deeply. Besides, Line 327-328 should be moved Section 1 or section4.

Response: The summary and the discussion have been reorganized. The summary now follows the discussion.

The sentence in Line 327-328 (now 292-294) is basis of the first part of our discussion, which was found by Wang et al. (2015). To keep the flow of argumentation tight we think it is appropriate at this place.

The main fining of this manuscript is that the GFDL-CM2p1 and MIROC5 exhibit better prediction skill of the EASM due to their ability in capturing the EASM-ENSO coupled mode.

Specific comments:

1. Comment: The conclusion of EASM-ENSO coupled mode should be put in the abstract.

Response: The EASM-ENSO coupled mode is defined by Wang et al. (2008). The different depiction of EASM-ENSO in the CMIP5 models can explain the different response to the initialisation in the models (Abstract, line 21-24). We added a sentence to the abstract to clarify why the simulation of this mode is important (line 24-25).

2. Comment: Line 162-164, the authors should give the sample size in calculating ACC.

Response: In section 2.1, we describe the datasets which are used in our study. The temporal coverage of these data is satellite era (1979 to 2005). Therefore, the sample sizes are 26.

3. Comment: Apart from the analysis of each figure, the authors should also give a brief summary of each figure. For instance, after Line 187, what is the key findings of Figure 1 and 2.

Response: Line 166-167 shows the finding of figure 1.

We added two sentences to clarify the summary of figure 2, line 187-190.

We also show the summary for each figure in Section 6.

The logic and structure of our paper can be found in response of minor comment 2.

Please also note the supplement to this comment:
https://www.earth-syst-dynam-discuss.net/esd-2017-51/esd-2017-51-AC1-supplement.pdf
* * *
[Figure]

**Supplement:**

[revised manuscript text omitted]

---

## Referee Comment (RC2) · Anonymous Referee #1 · 17 Aug 2017

The authors have well adressed all my concerns and did good job in revision. I suggest that this manuscript to be accepted to publish.

---

## Referee Comment (RC4) · Anonymous Referee #2 · 12 Sep 2017

SUMMARY:

This study evaluates the previously found link between the East Asian summer monsoon (EASM) and ENSO in CMIP5 models. It is stated that initialization can in some models improve the connection between ENSO and the EASM, and thereby the prediction of the monsoon.

GENERAL ASSESSMENT:

This study addresses an important teleconnection and an important region that would strongly benefit from better predictions on seasonal timescales. It does however not become clear if this study is able to solve any of the raised issues, and it leaves many open questions as to the methods used and the improvements obtained. The methods

section has to be considerably improved. My biggest concern is that the results remain unclear to the point that it is difficult to judge if the paper is worthy of publication. It is e.g. unclear which start dates are used for the models, and if the study compares different start dates to each other, which would render the results not comparable and mostly useless. The paper often reads more like a thesis than a paper. It should be made sure to explain the terms only where the reader cannot be assumed to know them, and the methods should be much more clearly explained. Throughout the paper, several paragraphs are misplaced and should be moved to a different section – e.g. the summary should not introduce new figures, and the results should not have to introduce new methods. The summary could be better structured. The English has to be improved – in several places it is difficult to understand what the authors mean.

DETAILED COMMENTS:

Line 9: sub-seasonal predictability is never mentioned again in the paper, probably better to leave it out in the abstract

Line 15: the zonal wind where?

Lines 15/16 and 19 – 21 seem to contradict each other, please clarify

Line 25: to predict -> change to: of predicting

Line 25: initialization of what?

Line 32: residents -> people

Line 33: South Asian monsoon-SAM is confusing, I would suggest writing ": : :submonsoon systems, the South Asian monsoon (SAM) and the East Asian monsoon (EAM) (Wang: : :).

Line 37: improve English: : : :predictions: : : is: : :

Line 41: how so?

Line 44/45: there are not really two kinds of climate models, it should be described as some being atmosphere-only with surface forcing, with the others including an ocean, while the atmosphere model stays the same. The reader does not need 10 lines of explanation of what the difference between an AGCM and an AOGCM is.

Line 59: SST where?

Line 60: initialization of what?

Line 69: what is an SST indicator?

Line 75: improve English

Lines 88 – 89: why do only these systems provide data? how were the runs obtained? How about the entire CHFP database? Are these used or was there a different source? CHFP would also include consistent start dates.

Line 89: it is never mentioned which start dates are used here, it is only stated that several start dates exist for some of the models. Are you comparing the same start dates for all models? It does not seem so from the available data. Unfortunately, the table does not clarify this either. If different start dates are used, the comparison is likely useless and would have to be repeated with consistent start dates.

Line 89: What does un-initialized mean? Are the "uninitialized" runs computed for a certain time frame?

Line 91: "on each pre-year 1st November": improve English

Line 98: change to: The configurations of the six prediction systems are summarized in Table 2.

Line 114-116: how about comparing the mean state in the models first before having to include the mean state in the PCC? That would make things simpler and clearer.

Lines 118 – 120: this is not clear, improve English

Lines 130 – 140: a bit lengthy

Line 158: I don't understand this sentence

Lines 168: : :: this should not be a part of the results, this should come earlier

Lines 180-182: which season is depicted here? This will be crucial for understanding what the paper is trying to say

Line 191: unfortunately I don't know what a sandwich pattern is supposed to look like. Since these EOFs are talked about so much here, they should be shown in a figure. It should be made clear why it helps to compute EOFs for predictability.

Lines 203/206: what is ua? Zonal wind anomaly? Anomaly with respect to what?

Line 229: this should be part of the methods

Line 235: I don't see this in Fig. 5

Line 239: when is it initialized?

Line 245: why is the correlation between Nino3.4 and the SOI important for predictability of the monsoon? Explain!

Line 263: how is that defined? Which months are used?

Line 294: so why exactly is there an improvement from non-initialized to initialized models?

Line 301: you cannot introduce a new figure in the summary. This should be part of the results

Line 317: this should come much earlier

Line 358: what are the IRI models?

Line 359: most of the models used here are seasonal forecast models already, not sure what you are trying to say

Line 365: SST where?

Figure 3: should be better explained in the main text

---

## Author Response (AR1)

We thank reviewer 2 very much for her/his helpful comments. We took her/his remarks into account and improved the manuscript accordingly.

**General comment:**

**Comment:** This study addresses an important teleconnection and an important region that would strongly benefit from better predictions on seasonal timescales. It does however not become clear if this study is able to solve any of the raised issues, and it leaves many open questions as to the methods used and the improvements obtained.

**Answer:** The East Asian summer monsoon (EASM) is an important and complex climate system. It is composited by different components. How to predict the EASM is a long-standing issue for climate community. This manuscript tries to find the better predictor to predict the EASM and the best initialisation strategy in CMIP5 models for further prediction study. We firstly analysed the seasonal predictability of six variables which contribute to the EASM system. Our finding is that both the lower (850 hPa) and upper (200 hPa) level zonal winds show better prediction skill than monsoon precipitation and meridional winds. A predictor based on the zonal winds might be a good option for EASM prediction. According to previous studies, the Wang and Fan monsoon index outperforms the other monsoon indices in representing the EASM characteristics. Therefore, we chose the Wang and Fan index for further study.

Compared to the non-initialised simulations, the GFDL-CM2p1 and the MIROC5 add prediction skill in simulating the EASM index with initialisation, the BCC-CSM1-1, the CanCM4, and the MPI-ESM-LR change the skill insignificantly, and the HadCM3 indicates a decreased skill score. We tried to answer the different response to the initialisation in the prediction systems. Then, we analysed the principal components of the simulations. With initialisation, the prediction systems enhance the first leading mode which is the ENSO developing mode of the EASM. We further analysed the interaction between the EASM and ENSO (EASM-ENSO coupled mode) in both non-initialised and initialised simulations. The different depiction of EASM-ENSO coupled mode in the systems can explain their different response to the initialisation.

This is the logic and structure of the manuscript. We think that the reader can easily follow it.

**Comment:** The methods section has to be considerably improved. My biggest concern is that the results remain unclear to the point that it is difficult to judge if the paper is worthy of publication. It is e.g. unclear which start dates are used for the models, and if the study compares different start dates to each other, which would render the results not comparable and mostly useless.

**Answer:** We re-wrote the section 2.1. It gives a clearer information about the models and the analysed data.

We pointed out the study time period is 1979-2005 in both abstract (line 12) and section 2.1 (models and initialisation; line 87-104). As mentioned in section 2.1, the prediction systems employ different initialisation strategies, we selected the following months (up to 12 months) of each simulation for further analysis. It is easy to evaluate the seasonal prediction skill of the prediction systems under their initialisation strategy. Our findings indicate that, under current initialisation strategy, the GFDL-CM2p1 and the MIROC5 can be used for further seasonal prediction study of EASM.

**Comment:** The paper often reads more like a thesis than a paper. It should be made sure to explain the terms only where the reader cannot be assumed to know them, and the methods should be much more clearly explained. Throughout the paper, several paragraphs are misplaced and should be moved to a different section – e.g. the summary should not introduce new figures, and the results should not have to introduce new methods. The summary could be better structured. The English has to be improved – in several places it is difficult to understand what the authors mean.

**Answer:** Thanks for this useful comment. We changed the structure of the manuscript.
1) The introduction/selection of EASM index and the calculation of ENSO index have been moved to section 2.3 (EAST ASIAN MONSOON INDEX AND ENSO INDEX);
2) We added two figures, Figure 1 shows the predictability of six variables in the multi-model ensemble mean for both non-initialised and initialised experiment, and Figure 4 presents the observed first lead EOF mode of the six variables;
3) We deleted the citation of figures in summary;
4) We improved the English through the manuscript.

**Detailed Comments:**

**Comment** Line 9: sub-seasonal predictability is never mentioned again in the paper, probably better to leave it out in the abstract

**Answer:** Revised, line 9.

**Comment** Line 15: the zonal wind where?

**Answer:** Revised the sentence to *"We found that most prediction systems simulated zonal wind over 850 and 200 hPa were significantly improved in the initialised simulations ...."*, line15-16.

**Comment** Lines 15/16 and 19 – 21 seem to contradict each other, please clarify

**Answer:** Revised the sentence.

**Comment** Line 25: to predict -> change to: of predicting

**Answer:** Done, line 27.

**Comment** Line 25: initialization of what?

**Answer:** changed the sentence to *"...predicting the EASM on a seasonal time-scale under the current initialisation strategy."* Line 27.

**Comment** Line 32: residents -> people

**Answer:** Done, line 34.

**Comment** Line 33: South Asian monsoon-SAM is confusing, I would suggest writing ". . .submonsoon systems, the South Asian monsoon (SAM) and the East Asian monsoon (EAM) (Wang. . .).

**Answer:** Done, line 35-36.

**Comment** Line 37: improve English: . . .predictions. . . is. . .

**Answer:** Revised, line 39.

**Comment** Line 41: how so?

**Answer:** We added one sentence to clarify it, *"This method establishes an empirical equation between the EASM and climate index…."* Line 44.

**Comment** Line 44/45: there are not really two kinds of climate models, it should be described as some being atmosphere-only with surface forcing, with the others including an ocean, while the atmosphere model stays the same. The reader does not need 10 lines of explanation of what the difference between an AGCM and an AOGCM is.

**Answer:** we modified these sentence to:

*"Without initialisation, both the atmosphere general circulation models (AGCMs) and coupled atmosphere-ocean general circulation models (CGCMs) cannot predict the climate on seasonal time-scale (Goddard et al., 2001). Given initial condition, the AGCMs have ability to predict the climate, but show low skill in predicting the EASM (Wang et al., 2005;Barnston et al., 2010). Because the AGCMs fail to produce the correctly relationship between the EASM and sea surface temperature (SST) anomalies over the tropical western North Pacific, the South China Sea, and the Bay of Bengal (Wang et al., 2004;Wang et al., 2005). Therefore, the monsoon community endeavours to*

*predict the EASM with CGCMs (Wang et al., 2008a;Zhou et al., 2009;Kim et al., 2012;Jiang et al., 2013)." Line 47-56.*

**Comment** Line 59: SST where?

**Answer:** Changed this sentence to *"...depends on their ability to reproduce the air-sea coupled process (Kug et al., 2008) and the given initial condition (Wang et al., 2005)." Line 59-61.*

**Comment** Line 60: initialization of what?

**Answer:** Changed this sentence to *"...depends on their ability to reproduce the air-sea coupled process (Kug et al., 2008) and the given initial condition (Wang et al., 2005)." Line 59-61*

**Comment** Line 69: what is an SST indicator?

**Answer:** Revised it to *"... SST variation index (i.e., El Niño-Southern Oscillation-ENSO index)...". Line 70-71*

**Comment** Line 75: improve English

**Answer:** Revised the sentence to *"2. Can the CMIP5 models capture the dynamical link between the ENSO and EASM?" Line 76*

**Comment** Lines 88 – 89: why do only these systems provide data? how were the runs obtained? How about the entire CHFP database? Are these used or was there a different source? CHFP would also include consistent start dates.

**Answer:** This study assesses the seasonal prediction skill of EASM in CMIP5 models. The six models provide yearly initialised simulations. The other CMIP5 models only provide 5-years initialised simulations. These models cannot be used for seasonal prediction study. The detailed description about the CMIP5 experiment can be found at Earth System Grid Federation (ESGF), *Taylor et al.,* (2012) and *Meehl et al.* (2014).

**Comment** Line 89: it is never mentioned which start dates are used here, it is only stated that several start dates exist for some of the models. Are you comparing the same start dates for all models? It does not seem so from the available data. Unfortunately, the table does not clarify this either. If different start dates are used, the comparison is likely useless and would have to be repeated with consistent start dates.

**Answer:** We revised the expression of the section 2.1 (models and initialisation).

Following the prediction framework of CMIP5, the six prediction systems established their initialisation strategy. There is a little bit difference of initialisation strategy between the six prediction systems (Table 2 and section 2.1). The reference papers in Table 1 show that the six systems have ability to predict the global climate with the current initialisation strategy. Intercomparison the six models can conclude the useful initialisation strategy for seasonal prediction of the EASM. In fact, our finding indicates that the GFDL-CM2p1 and the MIROC5 can be used for further seasonal prediction study of the EASM with the current initial strategy. Therefore, we think that the hypothesis of the paper is correct.

**Comment** Line 89: What does un-initialized mean? Are the "uninitialized" runs computed for a certain time frame?

**Answer:** We added some sentences to explain it. Line 90-93

*"For non-initialised simulations, the models were forced by observed atmospheric composition changes (reflecting both anthropogenic and natural sources) and, for the first time, including the time-evolving land cover (Taylor et al., 2012)."*

**Comment** Line 91: "on each pre-year 1st November": improve English

**Answer:** We added the initialised date for each model in Table 2.

**Comment** Line 98: change to: The configurations of the six prediction systems are summarized in Table 2.

**Answer:** Done.

**Comment** Line 114-116: how about comparing the mean state in the models first before having to include the mean state in the PCC? That would make things simpler and clearer.

**Answer:** We added a figure to show the predictable area of multi-model ensemble mean in both non-initialised and initialised experiment. Now it makes things easy to follow.

**Comment** Lines 118 – 120: this is not clear, improve English

**Answer:** Sentences have been improved.

**Comment** Lines 130 – 140: a bit lengthy

**Answer:** Paragraph has been reformulated and shortened.

**Comment** Line 158: I don't understand this sentence

**Answer:** We changed the sentence to *"These variables have been widely used to calculate monsoon index (Wang et al., 2008b)." line 183-184*

**Comment** Lines 168. . .: this should not be a part of the results, this should come earlier

**Answer:** The paragraph has been moved to section 2.3. Line 114-132

**Comment** Lines 180-182: which season is depicted here? This will be crucial for understanding what the paper is trying to say

**Answer:** The paper is talking about seasonal prediction of the EASM. This sentence shows clearly as *"In non-initialised simulations, none of the models captures the observed EASM, as indicated by an insignificant ACC (Figure 3)." Line 200-201*

Then, in section 2.3, there is a sentence to clarify the monsoon index *"The June-July-August mean of WF-index is used to represent the EASM for further analysis in this study." Line 126-127*

**Comment** Line 191: unfortunately I don't know what a sandwich pattern is supposed to look like. Since these EOFs are talked about so much here, they should be shown in a figure. It should be made clear why it helps to compute EOFs for predictability.

**Answer:** We added Figure 4 to show the observed first leading EOF model of monsoon precipitation and general circulation, and the associated principal component.

**Comment** Lines 203/206: what is ua? Zonal wind anomaly? Anomaly with respect to what?

**Answer:** In table 3, there is a sentence to explain *"The abbreviation of the variables is followed to the guidelines of CMIP5."* More details can be found at http://cmip-pcmdi.llnl.gov/cmip5/docs/standard_output.pdf

**Comment** Line 229: this should be part of the methods

**Answer:** It has been moved to section 2.3.

**Comment** Line 235: I don't see this in Fig. 5

**Answer:** Revised.

**Comment** Line 239: when is it initialized?

**Answer:** Changed it to *"...in initialised experiments." Line 250*

**Comment** Line 245: why is the correlation between Nino3.4 and the SOI important for predictability of the monsoon? Explain!

**Answer:** We revised the sentence (line 259-262). There is a short discussion about the correlation between Niño3.4 and the SOI in Section 5 (line 287-296).

**Comment** Line 263: how is that defined? Which months are used?

**Answer:** Figure 8 shows clearly that the lead-lag correlation coefficients between the EASM index and ENSO indices. This is defined as the ESNO-EASM coupled mode in previous monsoon studies. Here, we take this definition and analyse it in the simulations. As we mentioned, the EASM index is defined as the summer season (June-July-August; JJA) mean of monsoon index. The seasonal mean of ENSO indices (Niño3.4, and SOI) is applied to calculate the lead-lag correlation.

**Comment** Line 294: so why exactly is there an improvement from non-initialized to initialized models?

**Answer:** In section 5 (Discussion), we made a short discussion about the relationship between Niño3.4 and the SOI represented in non-initialised and initialised simulations.

*"The change of the correlation between Niño3.4 and the SOI is insignificant from non-initialised to initialised simulations. We therefore conclude that the relationship between Niño3.4 and the SOI depends more on the model parameterisation than on the initial condition." Line 298-302.*

**Comment** Line 301: you cannot introduce a new figure in the summary. This should be part of the results

**Answer:** You are right. We deleted the citation of figure in the summary.

**Comment** Line 317: this should come much earlier

**Answer:** The initialisation strategies are present in Table 2. Here, we mention it to discuss how the models work with the given initial condition. Line 287-289

**Comment** Line 358: what are the IRI models?

**Answer:** We added four sentences to explain the IRI models and discuss their limitation for EASM prediction. The four sentences as:

*"There are 4 AGCMs contributing to the IRI prediction system, including ECHAM4.5, CCM3.6, COLA and GFDL-AM2p14. These models are forced by prescribed SST to*

*forecast the climate on seasonal time-scale. Barnston et al. (2010) found that the models show low prediction skill over East Asia. Therefore, the IRI prediction system cannot be used to predict the EASM." Line 328-332*

**Comment** Line 359: most of the models used here are seasonal forecast models already, not sure what you are trying to say

**Answer:** Exactly, the two models (ECMWF System and the NCEP CFS) have been developed as seasonal forecast system. However, both the two systems show low prediction skill of EASM (Jiang et al., 2013; Kim et al., 2012). The CMIP5 models demonstrate a comparable prediction skill of EASM as the two seasonal forecast systems. Therefore, the CMIP5 models have potential to be developed as application system for EASM seasonal prediction, especially the GFDL-CM2p1 and the MIROC5.

**Comment** Line 365: SST where?

**Answer:** As we discussed in Line 303-325, the prediction systems cannot represent the second EOF lead mode which is the Indo-western Pacific monsoon-ocean coupled mode even given initial condition. Therefore, the prediction skill of EASM might can be improved while the systems optimise the SST-EASM interaction.

**Comment** Figure 3: should be better explained in the main text

**Answer:** There is a paragraph to describe the figure, line 213-227

Seasonal Prediction Skill of East Asian Summer Monsoon in CMIP5-Models

*Bo Huang,* Ulrich Cubasch, Christopher Kadow*

*Institute of Meteorology, Freie Universität Berlin,*
*Carl-Heinrich-Becker-Weg 6-10, 12165 Berlin, Germany*

*Email: huangb@zedat.fu-berlin.de*

**ABSTRACT**

The East Asian summer monsoon (EASM) is an important part of the global climate system and plays a vital role in the Asian climate. Its seasonal predictability is a long-standing issue within the monsoon scientist community. In this study, we will analyse the seasonal (the leading time is at least six months) prediction skill of the EASM rainfall and its associated general circulation in non-initialised and initialised simulations for the years 1979-2005

[revised manuscript text omitted]

---

## Author Response (AR2)

We thank reviewer 2 very much for her/his helpful comments. We took her/his remarks into account and improved the manuscript accordingly.

**Comment:** The start date of the models is still not addressed. The timing of the initialization date within the seasonal cycle crucially matters for the prediction of climate phenomena, and different timings of the initializations render the different models not comparable. This is a crucial issue that has to be addressed in the paper. If the models are not initialized at the same time, they are not comparable.

**Answer:** We analysed the seasonal prediction skill of East Asian summer monsoon (EASM) in six CMIP5 models. These models follow the CMIP5 framework, but establish their own initialisation strategy. A detailed description of the initialisation strategies for the six CMIP5 models can be found in Table 2. These datasets have been widely used in climate prediction research (Meehl et al., 2009;Meehl and Teng, 2012;Meehl et al., 2014;Choi et al., 2016a;Choi et al., 2016b). As the ocean is driving the long-term prediction skill rather than the initial condition of the atmosphere, the timing of the initialization has to be considered in the time scale of the ocean circulation, i.e. years to decades. Therefore, on an ocean time scale, the initialization takes place with comparable timing and therefore the results are comparable. This is a new approach based on decadal prediction experiments, which deviates from the scores of other seasonal prediction experiments based on initialization techniques derived from weather forecasting.

We added a paragraph to clarify our approach (Line 102-110).

**Comment:** My comment about CHFP has not been addressed at all. These models provide a large range of models that could be used to address the issues raised in this study for a large number of models with the same timing of the initialization.

**Answer:** Indeed, the CHFP provides a large range of forecast dataset for study the monsoon predictability based on the old seasonal prediction approach. The main goal of our paper is the comparison the seasonal prediction skill of EASM in CMIP5 models to investigate the potential of seasonal prediction using the decadal prediction methods. A comparison with published result can be found in the manuscript. A comprehensive comparison of all the CHFP data with the CMIP5 simulations regard to the seasonal prediction skill of the EASM is certainly an interesting topic, which should be addressed in an additional paper.

A sentence has been added to the text to clarify this issue (Line 345-352).

**Minor comments:**

**Answer:** we accordingly revised the manuscript by the comments.

Choi, J., Son, S. W., Ham, Y. G., Lee, J. Y., and Kim, H. M.: Seasonal-to-Interannual Prediction Skills of Near-Surface Air Temperature in the CMIP5 Decadal Hindcast Experiments, J Clim, 29, 1511-1527, 10.1175/Jcli-D-15-0182.1, 2016a.

Choi, J., Son, S. W., Seo, K. H., Lee, J. Y., and Kang, H. S.: Potential for long-lead prediction of the western North Pacific monsoon circulation beyond seasonal time scales, Geophys Res Lett, 43, 1736-1743, doi: 10.1002/2016GL067902, 2016b.

Meehl, G. A., Goddard, L., Murphy, J., Stouffer, R. J., Boer, G., Danabasoglu, G., Dixon, K., Giorgetta, M. A., Greene, A. M., Hawkins, E., Hegerl, G., Karoly, D., Keenlyside, N., Kimoto, M., Kirtman, B., Navarra, A., Pulwarty, R., Smith, D., Stammer, D., and Stockdale, T.: DECADAL PREDICTION Can It Be Skillful?, Bull Am Meteorol Soc, 90, 1467-1485, 10.1175/2009bams2778.1, 2009.

Meehl, G. A., and Teng, H. Y.: Case studies for initialized decadal hindcasts and predictions for the Pacific region, Geophys Res Lett, 39, L22705, 10.1029/2012gl053423, 2012.

Meehl, G. A., Goddard, L., Boer, G., Burgman, R., Branstator, G., Cassou, C., Corti, S., Danabasoglu, G., Doblas-Reyes, F., Hawkins, E., Karspeck, A., Kimoto, M., Kumar, A., Matei, D., Mignot, J., Msadek, R., Navarra, A., Pohlmann, H., Rienecker, M., Rosati, T., Schneider, E., Smith, D., Sutton, R., Teng, H. Y., van Oldenborgh, G. J., Vecchi, G., and Yeager, S.: DECADAL CLIMATE PREDICTION An Update from the Trenches, Bull Am Meteorol Soc, 95, 243-267, 10.1175/Bams-D-12-00241.1, 2014.

---

## Author Response (AR3)

Dear Prof. Valerio Lucarini,

Regarding to the initial date of the six prediction systems, we present the details of initial strategy in Table 2. We add a sentence in the caption of Table 2 to clarify the initial date (line 630). The six prediction systems contribute to CMIP5 project, which have been widely used to study climate prediction skill on seasonal-to-decadal scale (Choi et al. 2016; Meehl and Teng 2012; Meehl et al. 2014). Therefore, we employ the six prediction systems to discuss the seasonal prediction skill of east Asian summer monsoon. The six systems have performed a yearly initialisation (line 88-91). The initial date of every prediction year for each prediction system is shown in Table 2. A paragraph in Section 2.1 clarifies why the six prediction systems can be used to study prediction skill on seasonal time-scale (line103-111).

We expect you are agree with our clarification. Thank you very much.

Best regards,

Bo Huang

[revised manuscript text omitted]